# ContextFlow++: Generalist-Specialist Flow-based Generative Models with Mixed-Variable Context Encoding

**Denis Gudovskiy**[1]     **Tomoyuki Okuno**[2]     **Yohei Nakata**[2]

[1]Panasonic AI Lab, Mountain View, CA, USA
[2]Panasonic Holdings Corporation, Osaka, Japan

## Abstract

Normalizing flow-based generative models have been widely used in applications where the exact density estimation is of major importance. Recent research proposes numerous methods to improve their expressivity. However, conditioning on a context is largely overlooked area in the bijective flow research. Conventional conditioning with the vector concatenation is limited to only a few flow types. More importantly, this approach cannot support a practical setup where a set of context-conditioned (*specialist*) models are trained with the fixed pretrained general-knowledge (*generalist*) model. We propose ContextFlow++ approach to overcome these limitations using an additive conditioning with explicit generalist-specialist knowledge decoupling. Furthermore, we support discrete contexts by the proposed mixed-variable architecture with context encoders. Particularly, our context encoder for discrete variables is a surjective flow from which the context-conditioned continuous variables are sampled. Our experiments on rotated MNIST-R, corrupted CIFAR-10C, real-world ATM predictive maintenance and SMAP unsupervised anomaly detection benchmarks show that the proposed ContextFlow++ offers faster stable training and achieves higher performance metrics. Our code is publicly available at github.com/gudovskiy/contextflow.

## 1 INTRODUCTION

Recently, probabilistic generative models [Kingma et al., 2014] have gained attention as a solution for challenges in many fields e.g., molecular discovery [Bilodeau et al., 2022] and high-resolution image synthesis [Rombach et al., 2022]. An important class of such models are the bijective normaliz-

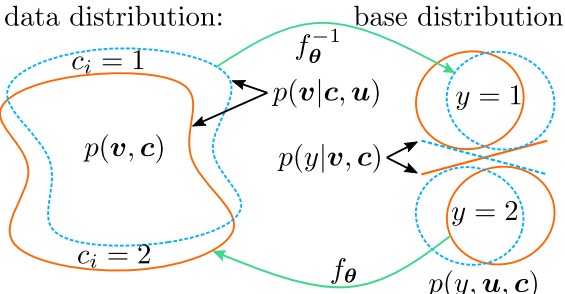

Figure 1: Normalizing flows implement a layered bijective transformations $f_{\theta_l}$ between a target data $p(\boldsymbol{v})$ distribution and a base $p(\boldsymbol{u})$ distribution using learned parameters $\boldsymbol{\theta}_l$. A trained model $f_{\boldsymbol{\theta}}$ usually predicts an outcome $p_{\boldsymbol{\theta}}(y|\boldsymbol{v})$ (right) or samples data using the learned $p_{\boldsymbol{\theta}}(\boldsymbol{v}|\boldsymbol{u})$ (left). When additional conditioning is needed to model $p(\boldsymbol{v}, \boldsymbol{c})$, the conventional approach with concatenated vectors $[\boldsymbol{v}_l, \boldsymbol{c}]$ is limited in the type of supported bijections and lacks the support of *generalist-specialist* training setup.

ing flows. Unlike variational autoencoders (VAEs) [Kingma and Welling, 2013] and diffusion models [Sohl-Dickstein et al., 2015], normalizing flows can estimate data likelihoods exactly. Therefore, flows are widely-used in semi-supervised prediction [Izmailov et al., 2020], time series forecasting [Rasul et al., 2021], unsupervised anomaly detection in computer vision [Gudovskiy et al., 2022], molecular graph generation [Kuznetsov and Polykovskiy, 2021] etc.

Current research on normalizing flows mostly aims to improve their likelihood estimation and sampling in various data domains [Kobyzev et al., 2020]. However, the conditioning in these models is a largely overlooked area. In particular, the conditioning is typically limited to concatenation of input data and context vectors [Lu and Huang, 2020]. At the same time, recent works on diffusion models show benefits of a more sophisticated ControlNet-style context-conditioning [Zhang et al., 2023] with the *generalist-specialist* setup.

Table 1: Previous (top) and the proposed (bottom) conditional bijections. A conditioning network (CN) processes contexts $c$. Previous methods either concatenate CN outputs with the internal RealNVP vectors for neural network (NN) processing or use only CN outputs in bijections. ContextFlow decouples CN and NN outputs using the additive operation while preserving bijection property. Symbols indicate: $\odot$ for element-wise multiplication and $\oslash$ for the division, ✓ for "yes" and ✗ for "no".

| Conditional Transformation | Inverse $f^{-1}: v, c \rightarrow u$ | Forward $f: u, c \rightarrow v$ | Bijective | Generalist -specialist |
|---|---|---|---|---|
| RealNVP coupling [Winkler et al., 2019], [Ardizzone et al., 2019] | $[v_a, v_b] = \text{SPLIT}(v)$ 
 $[s, t] = \text{NN}([v_b, \text{CN}(c)])$ 
 $u = [s \odot v_a + t, v_b]$ | $[u_a, u_b] = \text{SPLIT}(u)$ 
 $[s, t] = \text{NN}([u_b, \text{CN}(c)])$ 
 $v = [(u_a - t) \oslash s, u_b]$ | ✓ | ✗ |
| Actnorm [Lu and Huang, 2020] | $[s, t] = \text{SPLIT}(\text{CN}(c))$ 
 $\forall i, j: u_{i,j} = s \odot v_{i,j} + t$ | $[s, t] = \text{SPLIT}(\text{CN}(c))$ 
 $\forall i, j: v_{i,j} = (u_{i,j} - t) \oslash s$ | ✓ | ✗ |
| $\text{Conv}_{1 \times 1}^{-1}$ [Lu and Huang, 2020] | $W_c = \text{CN}(c)$ 
 $\forall i, j: u_{i,j} = W_c v_{i,j}$ | $W_c = \text{CN}(c)$ 
 $\forall i, j: v_{i,j} = W_c^{-1} u_{i,j}$ | ✓ | ✗ |
| RealNVP coupling (ours) | $[v_a, v_b] = \text{SPLIT}(v)$ 
 $[s, t] = \text{NN}(v_b) + \text{CN}(c)$ 
 $u = [s \odot v_a + t, v_b]$ | $[u_a, u_b] = \text{SPLIT}(u)$ 
 $[s, t] = \text{NN}(u_b) + \text{CN}(c)$ 
 $v = [(u_a - t) \oslash s, u_b]$ | ✓ | ✓ |
| Actnorm (ours) | $[s, t] = \text{SPLIT}([s, t]_v + \text{CN}(c))$ 
 $\forall i, j: u_{i,j} = s \odot v_{i,j} + t$ | $[s, t] = \text{SPLIT}([s, t]_v + \text{CN}(c))$ 
 $\forall i, j: v_{i,j} = (u_{i,j} - t) \oslash s$ | ✓ | ✓ |
| $\text{Conv}_{1 \times 1}^{-1}$ (ours) | $W_{g,c} = W_g + W_{\text{CN}}(c)$ 
 $\forall i, j: u_{i,j} = W_{g,c} v_{i,j}$ | $W_{g,c} = W_g + W_{\text{CN}}(c)$ 
 $\forall i, j: v_{i,j} = W_{g,c}^{-1} u_{i,j}$ | ✓ | ✓ |

Let's consider a practical setup in Figure 1 where a *generalist* model $f_\theta$ is trained on large-scale data $v$ that incorporates general knowledge about the data distribution $p(v)$. Assume a task to implement a probabilistic classifier $p_\theta(y|v, c)$ or a conditional sampling task $p_\theta(v|u, c)$, where there is a context $c$ that incorporates an additional context-specific knowledge. Then, a set of *specialists* can be learned using small-scale data from empirical distribution $p(v, c)$.

Conventional approach [Winkler et al., 2019] concatenates intermediate representations and the context $c$ inside the RealNVP coupling blocks [Dinh et al., 2017]. This preserves RealNVP invertability, but limits the type of supported bijections. In addition, this method cannot fully support a generalist-specialist setup where the context vector $c$ is missing at the generalist learning phase and it is introduced only later for domain-specific specialist training. Hence, it is unable to explicitly decouple the general and domain knowledge that is desired for complexity optimizations [Hu et al., 2022] and practical applications [Zhang et al., 2023].

To address the above limitations, we approach the tasks in Figure 1 setup as follows. First, a generalist model is trained with large-scale dataset to approximate $p(v)$ *without a-priori assumptions on the conditioning context*. Second, a set of specialist models with the defined context representations are learned with the *fixed generalist parameters* using small-scale training sets for each specialist. Hence,

we explicitly *decouple the generalist knowledge and a set of context-specific specialists* in the proposed ContextFlow++ model. Practical contexts $c$ are usually represented by discrete or mixed-precision variables that poses a difficulty because conventional flow framework supports only continuous variables. To overcome this difficulty, we use either embedding-based or variational dequantization methods implemented as sampling from an introduced encoding flow model. In summary, our contributions are as follows:

- We propose general approach to support additive context-conditioning for the generalist-specialist setup in bijective normalizing flow transformations.
- We address mixed-variable input data and contexts that are common in practical applications using the proposed ContextFlow++ architecture.
- Experiments show advantages of our ContextFlow++ approach in image classification, time series predictive maintenance and unsupervised anomaly detection.

## 2 RELATED WORK

**Normalizing flow architectures.** Normalizing flows [Kobyzev et al., 2020, Papamakarios et al., 2021] largely develop in a direction of increasing their expressivity and, hence, improving density estimation or sampling. For example, recent works propose continuous flows with the map-

pings obtained by solving neural ordinary differential equation (ODE) [Chen et al., 2018, Grathwohl et al., 2019] or process data with the manifold assumptions [Postels et al., 2022, Cunningham et al., 2022] or both [Chen and Lipman, 2024]. Our work is complementary to these more advanced models since we only consider a problem of context-conditioning.

The Flow++ architecture [Ho et al., 2019] proposes bijective transformation that models cumulative distribution function of a mixture with the fixed number of components. Each component is a distribution parameterized by neural network outputs. Such approach is partially related to ours because it implicitly models a mixture of densities at each bijection. Another related work proposes semi-supervised learning setup using a latent Gaussian mixture (FlowGMM) [Izmailov et al., 2020]. We employ FlowGMM-type method to predict an outcome $p(y|\boldsymbol{v}, \boldsymbol{c})$ for a discrete class $y$ that is independent of our context modeling goal.

**Context-conditioned flows.** In practice, the conditioning is very important feature in normalizing flow models, but research has been scarce in that area. The seminal works [Winkler et al., 2019, Ardizzone et al., 2019] propose to concatenate internal representations for a specific type of flow bijections i.e. the RealNVP couplings [Dinh et al., 2017] with the invertible conditioning. Lu and Huang [2020] extends conditioning to Glow bijections [Kingma and Dhariwal, 2018] i.e. the conditional actnorm and $\text{Conv}_{1\times1}^{-1}$ layers, where conditioning is performed by a separate discriminative neural network applied to context vector.

The above approaches have been widely adopted in many popular applications. For example, super-resolution images with rescaling can be generated using hierarchical conditional flow model [Liang et al., 2021] with feature-extracted context vectors. Unsupervised anomaly detection with segmentation can be improved by conditioning using positional encoding vectors [Gudovskiy et al., 2022]. Time series forecasting is performed by conditioning flow model on the outputs of a recurrent network [Rasul et al., 2021] or by multi-scale transformer-based attention with positional encoding [Feng et al., 2023]. Because the conventional conditioning methods cannot support generalist-specialist setup, we aim to introduce a *generic and principled alternative* to the effective yet limited concatenation-style conditioning.

**Discrete distribution modeling.** In Figure 1 we can have two distinct cases: the context vector $\boldsymbol{c}$ is represented by continuous variables or discrete variables. The former can be directly supported by the ContextFlow conditioning. However, the latter is more common in practice and requires additional (ContextFlow++) processing. Discrete densities can be converted to continuous ones by adding noise [Uria et al., 2013, Theis et al., 2016] or by using variational dequantization [Ho et al., 2019]. The $\arg\max$ variational method [Hoogeboom et al., 2021] additionally compresses discrete variables. Recent Voronoi dequantization [Chen et al., 2022]

learns quantization boundaries with an exact likelihood.

Another line of research processes discrete-only variables [Tran et al., 2019] and models continuous to discrete mappings [Sidheekh et al., 2022]. In this paper, we support mixed-variable contexts by the conventional flow framework, where categorical variables are mapped to continuous ones and added to the overall context. More complex contexts such as relational graphs with linked discrete variables [Fey et al., 2023] can be an avenue for future research.

**Discrete representations in other models.** In contrast to the specific task of discrete distribution modeling by normalizing flows in this paper, a general topic of using discrete representations has been pioneered by Bengio et al. [2013]. It introduces the straight-through gradient estimator to learn discrete (quantized) representations in discriminative models. Then, the vector-quantized VAE (VQ-VAE) [van den Oord et al., 2017] with discretized encoder's latent space employs such estimator to avoid posterior collapse in the generative VAE model. Later, this approach has been widely applied to other generative models such as generative adversarial networks [Esser et al., 2021] and diffusion models [Hoogeboom et al., 2021]. Recent methods [Bond-Taylor et al., 2022, Chang et al., 2022] rely on the transformer architecture to learn a codebook that is indexed by discrete indices, where it implements controllable data (e.g., image) synthesis and manipulations. Though recent continuous normalizing flows [Lipman et al., 2023] can compete with the diffusion models in synthesis, we, for simplicity, consider a task of mixed-variable density estimation using finite normalizing flow architectures in this paper. In that task, VAEs and diffusion models are unable to estimate exact data likelihoods even for continuous data variables.

# 3 PRELIMINARIES

## 3.1 NORMALIZING FLOW FRAMEWORK

Normalizing flows [Rezende and Mohamed, 2015] can transform a target density $p_V$ of data vectors $\boldsymbol{v} \in \mathcal{V} = \mathbb{R}^D$ to a base density $p_U$ with vectors $\boldsymbol{u} \in \mathcal{U} = \mathbb{R}^D$ using the change-of-variable formula by *bijective* and *differentiable* transformation $f : \mathcal{U} \to \mathcal{V}$ at any point as

$$p_V(\boldsymbol{v}) = p_U(\boldsymbol{u}) \left| \det \partial \boldsymbol{u}/\partial \boldsymbol{v}^T \right|, \text{ and } \boldsymbol{u} = f_{\boldsymbol{\theta}}^{-1}(\boldsymbol{v}), \quad (1)$$

where a base random variable $\boldsymbol{u}$ can be from a standard Gaussian or a parameterized distribution. The normalizing flow model $f_{\boldsymbol{\theta}}$ with $\boldsymbol{\theta}$ parameters is typically implemented as a sequence of tractable transformations.

The closed-form expression in (1) allows to learn exact density functions. However, this conventional flow framework is restricted to only *continuous* $p_V(\boldsymbol{v})$ densities.

## 3.2 DEQUANTIZATION METHODS

To learn densities $P(\boldsymbol{x})$ of *discrete* variables, it is common to apply a *surjective* [Nielsen et al., 2020] transformation $g : \mathcal{X} \to \mathcal{V}$ that is deterministic in one direction ($\boldsymbol{x} = g_{\boldsymbol{\lambda}}^{-1}(\boldsymbol{v})$) and stochastic in the other ($\boldsymbol{v} \sim q_{\boldsymbol{\lambda}}(\boldsymbol{v}|\boldsymbol{x})$) using a dequantization distribution $q_{\boldsymbol{\lambda}}(\boldsymbol{v}|\boldsymbol{x})$ with parameters $\boldsymbol{\lambda}$ i.e. the dequantization model. Then, the discrete density can be written using Dirac $\delta$-function as

$$P(\boldsymbol{x}) = \int P(\boldsymbol{x}|\boldsymbol{v})p(\boldsymbol{v})d\boldsymbol{v}, P(\boldsymbol{x}|\boldsymbol{v}) = \delta(\boldsymbol{x} = g_{\boldsymbol{\lambda}}^{-1}(\boldsymbol{v})). \quad (2)$$

A *surjective* encoder $g_{\boldsymbol{\lambda}}(\boldsymbol{x})$ in (2) estimates the evidence lower bound (ELBO) of $q_{\boldsymbol{\lambda}}(\boldsymbol{v}|\boldsymbol{x})$ as

$$\log P_{\boldsymbol{\lambda}}(\boldsymbol{x}) \geq \mathbb{E}_{\boldsymbol{v} \sim q_{\boldsymbol{\lambda}}(\boldsymbol{v}|\boldsymbol{x})} \left[ \log p(\boldsymbol{v}) - \log q_{\boldsymbol{\lambda}}(\boldsymbol{v}|\boldsymbol{x}) \right], \quad (3)$$

where the ELBO holds for the support $\mathcal{S} = \{\boldsymbol{v} \in \mathbb{R}^D : \boldsymbol{x} = g_{\boldsymbol{\lambda}}^{-1}(\boldsymbol{v})\}$ such that $P(\boldsymbol{x}|\boldsymbol{v}) = 1$ in that support region.

Typical surjection choice is the rounding operation $\lfloor \boldsymbol{v} \rfloor$ [Uria et al., 2013, Theis et al., 2016, Ho et al., 2019], $\arg\max(\boldsymbol{v})$ operation [Hoogeboom et al., 2021] or a set identification function $\mathbb{R}^D \to \{1, \dots, K\}$ [Chen et al., 2022].

Then, *discrete variables* can be processed using the (2,3) generic framework by choosing the appropriate $g_{\boldsymbol{\lambda}}$ with a corresponding dequantization model $q_{\boldsymbol{\lambda}}(\boldsymbol{v}|\boldsymbol{x})$ at the expense of the ELBO estimate rather than the exact likelihood. Dequantization model can simply add noise from uniform distribution [Uria et al., 2013, Theis et al., 2016], or samples from a flow model that implements variational distribution $q_{\boldsymbol{\lambda}}(\boldsymbol{v}|\boldsymbol{x})$ [Ho et al., 2019, Hoogeboom et al., 2021]. Unlike the variational approaches with the ELBO estimate of discrete density, the use of disjoint subsets provides an exact likelihood [Chen et al., 2022].

## 3.3 CONVENTIONAL CONDITIONAL FLOWS

In the conditional setting, there is an additional context vector $\boldsymbol{c} \in \mathcal{C}$ with the $p_C$ density. Often, the context is given by discrete variables which can be addressed by Section 3.2 methods and the proposed in Section 4 framework.

Then, assuming continuous data and context vectors, we can rewrite (1) for the joint log-likelihood as

$$\log p_{\boldsymbol{\theta}}(\boldsymbol{v}, \boldsymbol{c}) = \log p_{\boldsymbol{\gamma}}(\boldsymbol{u}) + \sum\nolimits_{l=1}^{L} \log |\det \boldsymbol{J}_l|, \quad (4)$$

where $\boldsymbol{u} = f_{\boldsymbol{\theta}}^{-1}(\boldsymbol{v}; \boldsymbol{c})$, the Jacobian matrices $\boldsymbol{J}_l$ are sequentially calculated for the $l^{\text{th}}$ transformation $f_{\boldsymbol{\theta}_l}^{-1}$, and $\boldsymbol{\gamma}$ represents parameters of the base distribution.

Next, we formally summarize previously proposed conditional bijections for (4) in Table 1 (top). Currently, they are limited to RealNVP coupling bijections, Glow's activation

normalization and $\text{Conv}_{1 \times 1}^{-1}$ layers. A conditional neural network (CN) with the context input estimates bijection parameters. These parameters are either directly used (Actnorm and $\text{Conv}_{1 \times 1}^{-1}$) or concatenated with the intermediate vectors (RealNVP) to calculate bijection's output.

Conventional methods are viable when assumptions about $\boldsymbol{c}$ are known and there is access to $p(\boldsymbol{v}, \boldsymbol{c})$ data. However, often practitioners employ context information after learning on $p(\boldsymbol{v})$ data. For example, a pretraining step with large-scale data can be performed to extract *general* knowledge followed by context-conditional *specialist* training with fixed generalist parameters. Moreover, in some situations we cannot anticipate what type of context information can be useful for a task or metric. Hence, we propose a framework where data and context modeling is explicitly decoupled.

# 4 PROPOSED METHOD

## 4.1 ADDITIVE CONTEXT FOR SPECIALISTS

Let's consider a modified setup for the (4) task, where the generalist model $f_{\boldsymbol{\theta}_g}^{-1}$ is pretrained using the conventional objective (1) to estimate $\log p_{\boldsymbol{\theta}_g}(\boldsymbol{v})$. We are interested in improving the generalist likelihood estimates for each specific context without modifying its $\boldsymbol{\theta}_g$ parameters.

With the exception of masked autoregressive flows [Papamakarios et al., 2017], it is common to model elements $v_i$ of an input data vector $\boldsymbol{v} = [v_1, \dots v_D]$ as independent variables in a single bijection layer [Papamakarios et al., 2021]. Then, it is presumed that a sufficient number of bijection layers with $v_i$ permutations lead to an accurate joint likelihood estimate for $\boldsymbol{v}$. Similarly, we can assume that the context vector $\boldsymbol{c}$ is independent of data vector $\boldsymbol{v}$ within a bijection and can rewrite joint log-likelihood (4) as a sum

$$\log p_{\boldsymbol{\theta}_{g,c}}(\boldsymbol{v}, \boldsymbol{c}) = \log p_{\boldsymbol{\theta}_g}(\boldsymbol{v}) + \log p_{\boldsymbol{\theta}_c}(\boldsymbol{c}), \quad (5)$$

where the generalist parameters $\boldsymbol{\theta}_g$ are fixed after the pretraining and only the specialist parameters $\boldsymbol{\theta}_c$ are learned.

With the (5) assumption for (4), we propose a density transformation approach with additive log-likelihood contributions. In particular, consider a generalized bijection

$$\boldsymbol{u} = \boldsymbol{M} \left( \boldsymbol{W}_g + \boldsymbol{W}_c \right) \boldsymbol{v}, \; \boldsymbol{v} = \left( \boldsymbol{M} \left( \boldsymbol{W}_g + \boldsymbol{W}_c \right) \right)^{-1} \boldsymbol{u}, \; (6)$$

where matrices $\boldsymbol{W}_{g,c}$ are parameters for any flow type [Kobyzev et al., 2020] e.g., element-wise or linear. RealNVP and autoregressive couplings with appropriate binary mask $\boldsymbol{M}$ can be implemented with $\boldsymbol{W}_g = \text{NN}(\boldsymbol{v})$ and $\boldsymbol{W}_c = \text{CN}(\boldsymbol{c})$. Since (6) implements a linear combination, its Jacobian terms are additive $\partial \boldsymbol{u} / \partial \boldsymbol{v}^T = \boldsymbol{M}\boldsymbol{W}_g + \boldsymbol{M}\boldsymbol{W}_c$ with decoupled likelihood contributions $|\det \boldsymbol{J}_g|$ and $|\det \boldsymbol{J}_c|$.

We apply our additive context approach to several common bijections using (6). Table 1 (bottom) contains examples

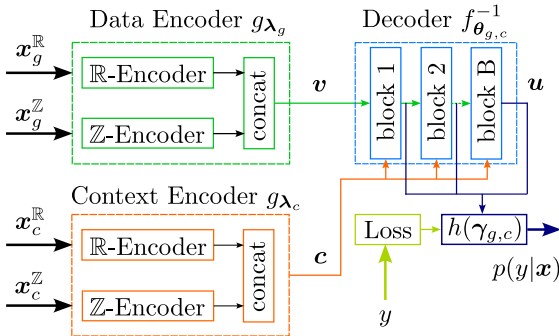

Figure 2: Our high-level scheme. Mixed-variable inputs and contexts are represented by vectors $x_{g,c}^{\mathbb{R},\mathbb{Z}}$. First, the data encoder $g_{\lambda_g}$ and decoder $f_{\theta_g}^{-1}$ are learned during large-scale generalist pretraining step. Next, the specialist context encoder $g_{\lambda_c}$ and the extended decoder parameters $f_{\theta_c}^{-1}$ are learned with small-scale data. Generative encoders convert discrete variables into continuous data $v$ and context $v$ vectors. A distributional model $h(\gamma_{g,c})$ also supports such two-step training and outputs likelihood $p(y|x)$ estimates.

for Glow-type finite flows. The main difference between previous and our transformations is the *explicit separation of generalist and specialist processing*.

## 4.2 ENCODING MIXED-VARIABLE DATA

As discussed in Section 3.2 input data can be heterogeneous i.e. represented by continuous ($x^{\mathbb{R}} \in \mathbb{R}$) or discrete ($x^{\mathbb{Z}} \in \mathbb{Z}$) variables. This is especially relevant to the context vectors which often contain information represented by integers such as user preferences, sensor configurations, product's geographical location etc. Therefore, we propose to extend conventional normalizing flow architecture (bijective decoder) by an additional encoding step as shown in Figure 2 scheme. The encoding step for continuous inputs is optional and can contain data preprocessing e.g., normalization. However, it is essential for discrete inputs to use either embedding-based [Gorishniy et al., 2022] mappings or one of the dequantization methods from Section 3.2.

In this paper, we also investigate trade-offs of various methods for mapping discrete variables to continuous ones when applied to probabilistic flow framework. First, a common practical solution is to use a differentiable embedding to look up a learnable continuous-space vector by a discrete index. There are many variants of this approach as described in [Gorishniy et al., 2022]. This simple deterministic lookup method can be extended to a stochastic sampling from a learnable distribution $q_{\lambda}(v|x)$. The latter can be seen as a special case of dequantization.

Second, low-complexity uniform [Uria et al., 2013, Theis et al., 2016] dequantization is a popular choice for images and audio sequences. It typically works well for discrete data

with relatively high cardinality (e.g., 8-bit variables have cardinality of 256). Thus, we adopt uniform dequantization in our experiments for input data to avoid high complexity.

Variational dequantization methods with rounding [Ho et al., 2019] and $\arg\max$ operations [Hoogeboom et al., 2021] offer more accurate parametric mappings for low-cardinality categorical data. Let the $i^{\text{th}}$ variable $x_i^{\mathbb{Z}} \in \mathbb{Z}_i = \{1, 2, \ldots, K_i\}^D$ represent a discrete vector with $K_i$ categories. If drop the index $i$ for convenience, $x$ is the input to $g_{\lambda}$ parametric encoder. The encoder implements a *surjective* mapping $g : \mathbb{Z} \to \mathbb{R}$ between discrete $x$ and continuous $v$. Then, our encoder outputs $v \in \mathbb{R}^D$ for variational dequantization and $v \in \mathbb{R}^{D \times K}$ for the $\arg\max$ method.

Furthermore, we experiment with the variational method with rounding that maps $x$ to one-hot binary representation $v \in \mathbb{R}^{D \times K}$ [Gorishniy et al., 2022]. Additionally, a naïve implementation of the $\arg\max$ approach has significant complexity due to large $v \in \mathbb{R}^{D \times K}$ vectors for each $x^D$. To reduce complexity, we apply Cartesian product compression as in [Hoogeboom et al., 2021]. Then, the number of dimensions to encode each discrete variable is the lowest for $\log_2$ (binary) representation. This approach encodes categorical discrete variables to $v \in \mathbb{R}^{D \times \log_2 K}$ outputs.

To summarize, we propose a mixed-variable probabilistic architecture to support various kinds of input and context data with details shown in Figure 3. Our encoding step for discrete variables can be implemented with different types of vector mappings followed by several dequantization and embedding-based methods as described above. Effectively, we *generalize the encoder as a surjective normalizing flow model* using stochastic right inverse $g_{\lambda}$. Therefore, the encoder implements variational methods by sampling from the parameterized distribution $q_{\lambda}(v|x)$ and, additionally, can contain flow's transformations to be more expressive when generating continuous variables. At the same time, our ContextFlow++ decoder extends the conventional bijective flow model that performs deterministic inverse $f_{\theta}^{-1}$.

## 4.3 OVERALL ARCHITECTURE

Section 4.2 gives details about mapping and $q_{\lambda}(v|x)$ sampling variants in our context encoder presented in Figure 3. In addition, our context encoder as well as the decoder contains $B$ bijective blocks. Each block consists of a *squeeze* layer, a sequence of $L$ sub-blocks and an optional *split prior* layer [Nielsen et al., 2020]. The *squeeze* layer reduces each spatial or temporal dimension by a factor of 2 and, correspondingly, increases data dimensions. The split prior layer reduces data dimensions by a factor of two and applies a distributional model for a half of them, which is applied only to CIFAR-10C and ATM datasets in Section 5.

Each of $L$ sub-blocks contains our modified Glow-type transformations from Table 1. To be precise, sub-blocks

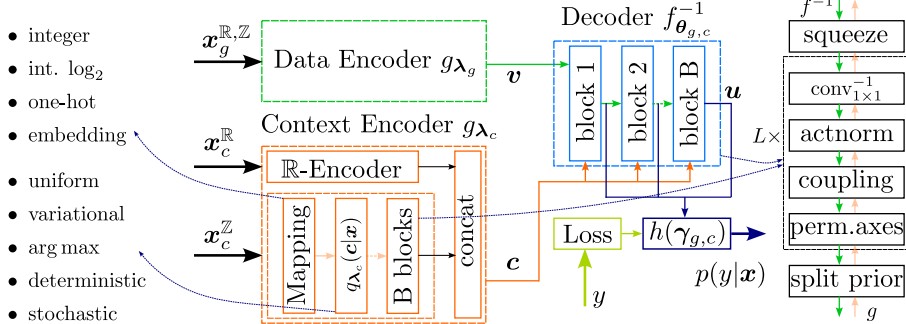

Figure 3: Our detailed ContextFlow++ architecture with mixed-variable data and context encoders $g_{\lambda_{g,c}}$ that are implemented as a sampling from the surjective flow model with various discrete-variable mapping and distribution options. The bijective flow decoder performs likelihood estimation using the encoder's $v$ input during generalist training step with $\theta_g$ parameters. Then, it is followed by context-specific specialist training with $\theta_c$ parameters using sampled contexts $c$. A distributional model $h(\gamma_{g,c})$ implements task's probabilistic classifier and outputs $p(y|x)$ likelihood estimates.

function as the conventional transformations at generalist training step. Then, the context-specific processing is added during specialist learning step, while the generalist parameters are fixed. Our neural networks in the coupling layers have an option to be convolutional or, optionally, have the ViT transformer [Dosovitskiy et al., 2021]. We also permute data and temporal axes for the time series ATM dataset in Section 5 experiments using the *permute axes* layer.

Our distributional model $h(\gamma_{g,c})$ with diagonal Gaussian base distribution implements the FlowGMM-style probabilistic classifier [Izmailov et al., 2020, Gudovskiy et al., 2023] with 8 mixture components and $M$ classes for each outcome $y = m$ $(m = 1 \ldots M)$ and a corresponding set of learnable parameters: means, variances and weights. It also supports separate modeling of generalist and context-specific distributions using two sets of the above parameters. The distributional model outputs $p(x, c|y)$ likelihood estimates that are used in the loss function. To support semi-supervised setting, we use the loss that consists of supervised cross-entropy and unsupervised terms expresses by

$$
\begin{aligned}
\mathcal{L} = -\frac{1}{|\mathbb{N}|} \sum_{i \in \mathbb{N}} [&\log \text{softmax} \, \log p(x_i, c_i|y_i = m) \\
&+ \alpha \log \sum_m p(x_i, c_i|y_i = m)],
\end{aligned} \quad (7)
$$

where $\mathbb{N}$ is the training set, $\text{softmax}$ computes classifier's predictions $p(y_i|x_i, c_i)$ and the hyperparameter $\alpha =$1e-3. The first term in (7) is omitted in the unsupervised experiments $(M = 1)$ and only the last term with $\alpha =$1 is retained.

# 5 EXPERIMENTS

## 5.1 EXPERIMENT SETUP

**Benchmarks.** Though recent continuous flows [Lipman et al., 2023] can compete with the diffusion models in

$p(x|u)$ sampling or can be a latent-space component in the sampling pipeline [Davtyan et al., 2023], we are mostly interested in modeling $p(y|x, c)$ predictions using well-established finite flow architectures [Kingma and Dhariwal, 2018] from Table 1. Particularly, we experiment with the discrete contexts and the generalist-specialist setup.

Hence, we select four benchmarks. First, we modify small-scale MNIST classification with $M = 10$ classes by applying $c \sim \mathcal{U}\{0, 63\}$ random image rotations with $360°/64$ discrete steps to all data splits. Such rotated MNIST-R defines a simple yet challenging task for conventional architectures without inherent rotational invariance property.

Second larger-scale image classification benchmark is the widely-used CIFAR-10C [Hendrycks and Dietterich, 2019] with synthetic corruptions. We define 2-dimensional context vector in CIFAR-10C as $c \sim [\mathcal{U}\{1, 15\}, \mathcal{U}\{1, 5\}]$ that models discretely sampled image corruption type (15) and its severity level (5), respectively. When applied to image classification, CIFAR-10C corruptions usually cause a significant drop in the prediction accuracy.

Lastly, we employ two real-world time series benchmarks: supervised ATM machine failure prediction [Vargas et al., 2023] and SMAP unsupervised anomaly detection Hundman et al. [2018]. ATM dataset contains 29,386 sequences collected from 68 deployed ATM machines, where each 144-length sequence has 38 data dimensions. The task is to predict an ATM failure in one-week time frame using binary labels $(M = 2)$. Then, we use ATM machine ID as a discrete context. Second, the soil moisture active passive satellite (SMAP) dataset contains soil samples and telemetry information from the Mars rover with 135,183 and 427,617 data points in the training (without anomalies $M = 1$) and test sets, respectively. SMAP data has 25 data dimensions collected from 55 entities. We use the entity ID as a discrete context for our ContextFlow++. We follow Su et al. [2019] and transform the regression task into a classification task

Table 2: Small-scale image classification benchmark using MNIST-R with 64 rotations. Each rotation represents a conditioning context. The **best** and the second best top-1 accuracy ($\mu_{\pm\sigma}$, %) results are highlighted. The generalist model experiences 2.8 p.p. accuracy drop when adding image rotations. The prior context-conditioned model [Lu and Huang, 2020] trained from scratch and our ContextFlow++ trained with the fixed generalist parameters show similar accuracy gains.

| Context Encoder → Model ↓ | Fixed Generalist | Integer | | One-hot binary | | Learned embedding | |
| --- | --- | --- | --- | --- | --- | --- | --- |
| | | uniform | arg max | uniform | variational | deterministic | stochastic |
| Generalist$_{\text{with rot.}}$ | | w/o → with rotations: $98.9_{\pm0.1} \rightarrow 96.1_{\pm0.2}$ (2.8 p.p. drop) | | | | | |
| Lu and Huang [2020] | ✗ | $97.6_{\pm0.1}$ | $97.7_{\pm0.1}$ | $97.4_{\pm0.1}$ | **$97.8_{\pm0.1}$** | $97.7_{\pm0.1}$ | **$97.8_{\pm0.1}$** |
| ContextFlow++ (ours) | ✓ | $97.7_{\pm0.1}$ | $97.8_{\pm0.1}$ | $97.6_{\pm0.1}$ | **$97.9_{\pm0.1}$** | **$97.9_{\pm0.1}$** | $97.8_{\pm0.1}$ |

Table 3: Larger-scale image classification benchmark using CIFAR-10C with corruptions. Corruption type and its severity define 2-dimensional conditioning context. The **best** and the second best top-1 accuracy ($\mu_{\pm\sigma}$, %) results are highlighted. The generalist model experiences 6.6 p.p. accuracy drop when adding image corruptions. The prior method [Lu and Huang, 2020] cannot surpass even the generalist results. Our ContextFlow++ with the fixed general knowledge show higher classification accuracy, in particular, with more advanced context encoders i.e. variational and embedding-based.

| Context Encoder → Model ↓ | Fixed Generalist | Integer | | One-hot binary | | Learned embedding | |
| --- | --- | --- | --- | --- | --- | --- | --- |
| | | uniform | arg max | uniform | variational | deterministic | stochastic |
| Generalist | | w/o → with corruptions: $61.7_{\pm1.3} \rightarrow 55.1_{\pm0.3}$ (6.6 p.p. drop) | | | | | |
| Lu and Huang [2020] | ✗ | $49.0_{\pm0.9}$ | $51.2_{\pm0.4}$ | $48.3_{\pm2.6}$ | $50.8_{\pm0.5}$ | **$52.5_{\pm0.4}$** | $50.8_{\pm0.7}$ |
| ContextFlow++ | ✓ | $56.5_{\pm0.3}$ | $57.1_{\pm0.4}$ | $57.3_{\pm0.4}$ | **$57.7_{\pm0.5}$** | $57.4_{\pm0.3}$ | $56.8_{\pm0.3}$ |

using sliding windows (window size = 8) and replication padding Tuli et al. [2022]. Both datasets are imbalanced with $\approx 10\%$ of positive (failure or anomaly) labels.

**Flow models.** We experiment with the Glow-type models from Table 1 with the following dequantization. First, we always apply low-complexity uniform dequantization to the $x$ inputs. Second, we employ Section 4.2 surjective context encoders for conditioning. Particularly, we experiment with the following context encoders: with uniform [Theis et al., 2016] and variational dequantization methods [Ho et al., 2019, Hoogeboom et al., 2021] as well as trainable embedding-based deterministic and stochastic encoders using the library from Gorishniy et al. [2022]. Also, we apply dequantization methods both to the original integer contexts and to their one-hot binary representations. Context representation is important due to computational complexity and dequantization considerations. For example, the $\arg\max$ method is only applicable to integer contexts, while variational [Ho et al., 2019] approach is well-suited for binary one-hot representation. We apply the same flow architecture in all benchmarks with variable number of blocks and sub-blocks as presented in Section 4.3. We select (number of blocks $B$, and sub-blocks $L$) as (2,2) for MNIST-R, (3,4) for CIFAR-10C and ATM, (2,4) for SMAP, respectively. We apply convolutional couplings in MNIST-/CIFAR-10C image classification datasets and transformer-based couplings in ATM/SMAP time series datasets.

**Training hyperparameters.** We train the generalist and the conventional specialist models [Lu and Huang, 2020] from the scratch for each benchmark. Then, we train our Con-

textFlow++ model with the pretrained generalist parameters. Since ContextFlow++ explicitly decouples the general and context-specific knowledge, there are two sets of parameters: one fixed set inherited from the generalist and a learnable set for additive conditioning in the context encoder.

Each model is optimized with the following hyperparameters: AdamW optimizer with 256-size batches and initial 1e-3 learning rate, which is reduced by a factor of 10 every 12 epochs with 48 epochs in total. A warm-up phase with the learning rate gradually increasing from 1e-4 to 1e-3 is applied during first 4 epochs.

**Evaluation.** We use top-1 accuracy metric for MNIST-R and CIFAR-10C classification tasks. In addition, the standardized metrics from Vargas et al. [2023] are used for ATM failure prediction: balanced and unbalanced top-1 accuracies, area under the receiver operating characteristic curve (AuROC), average precision (AP), $F_1$-score [Lipton et al., 2014] and minimum sensitivity (MS). We also rely on $F_1$-score to compute a binary prediction threshold in ATM. We follow Su et al. [2019] and report precision (P), recall (R), AuROC and $F_1$ score for the SMAP dataset.

We run each experiment four (MNIST-R, CIFAR-10C, SMAP) or five (ATM) times and report the metric's mean ($\mu$) and, if shown, standard deviation ($\pm\sigma$) on the test split. Unlike other datasets with the fixed training/test splits, we perform 5-fold cross-validation splits with a single seed (2) for ATM. Note that unlike [Vargas et al., 2023], we do not perform context-stratified splits to have overlapping contexts in training/test splits. The latter choice increases performance metrics that have been reported in their paper.

Table 4: Real-world ATM machine failure prediction with time series sensory data [Vargas et al., 2023]. Machine IDs define the conditioning context. The reference flow-based generalist model outperforms other baseline models. Our ContextFlow++ further improves performance metrics. The variational and deterministic embedding-based context encoders achieve the highest metrics. The **best** and the underline{second best} metric's ($\mu_{\pm\sigma}$, %) results are highlighted.

| Model Metric ↓ | random forests | HYDRA | XGBoost | Embed. determ. Lu and Huang [2020] | ContextFlows → Generalist Flow ↓ | Integer arg max | One-hot variational | Embed. determ. |
|---|---|---|---|---|---|---|---|---|
| Accuracy | $94.39_{\pm0.6}$ | $81.3_{\pm0.5}$ | $96.7_{\pm0.2}$ | $96.7_{\pm0.3}$ | $97.2_{\pm0.5}$ | $98.1_{\pm0.3}$ | $\underline{98.3_{\pm0.2}}$ | $\mathbf{98.5_{\pm0.1}}$ |
| Bal. Acc. | $73.71_{\pm3.0}$ | $74.1_{\pm0.4}$ | $85.3_{\pm1.2}$ | $91.9_{\pm1.6}$ | $91.5_{\pm1.6}$ | $\underline{95.1_{\pm1.1}}$ | $95.0_{\pm0.6}$ | $\mathbf{95.9_{\pm0.2}}$ |
| AuROC | $73.71_{\pm3.0}$ | $74.1_{\pm0.4}$ | $85.3_{\pm1.2}$ | $99.0_{\pm0.2}$ | $98.7_{\pm0.2}$ | $\underline{99.4_{\pm0.2}}$ | $\underline{99.4_{\pm0.2}}$ | $\mathbf{99.6_{\pm0.1}}$ |
| AP | $51.73_{\pm5.5}$ | $23.8_{\pm0.5}$ | $71.0_{\pm1.9}$ | $93.2_{\pm1.4}$ | $92.7_{\pm1.5}$ | $96.0_{\pm0.8}$ | $\underline{97.0_{\pm0.3}}$ | $\mathbf{97.1_{\pm0.5}}$ |
| $F_1$ | $63.68_{\pm5.3}$ | $42.0_{\pm0.7}$ | $81.5_{\pm1.5}$ | $84.3_{\pm1.6}$ | $86.3_{\pm2.7}$ | $90.9_{\pm1.4}$ | $\underline{91.9_{\pm0.8}}$ | $\mathbf{92.6_{\pm0.6}}$ |
| MS | $47.59_{\pm6.1}$ | $65.0_{\pm0.7}$ | $70.9_{\pm2.4}$ | $85.9_{\pm3.4}$ | $84.4_{\pm3.0}$ | $\underline{91.2_{\pm2.3}}$ | $90.9_{\pm1.4}$ | $\mathbf{92.6_{\pm0.4}}$ |

Table 5: Subsampled ATM machine failure prediction benchmark with the increased to $100\times$ positive/negative data imbalance. As a result, the performance gaps between ContextFlow++ specialists and other models are also increased. Unlike the previous setup, the variational context encoder outperforms the deterministic embedding-based encoder, which highlights advantages of a more robust fully-probabilistic approach in real-world applications.

| Model Metric ↓ | random forests | HYDRA | XGBoost | Embed. determ. Lu and Huang [2020] | ContextFlows → Generalist Flow ↓ | Integer arg max | One-hot variational | Embed. determ. |
|---|---|---|---|---|---|---|---|---|
| Accuracy | $91.56_{\pm0.4}$ | $78.5_{\pm1.6}$ | $\mathbf{93.0_{\pm0.2}}$ | $90.9_{\pm1.1}$ | $90.9_{\pm0.9}$ | $91.9_{\pm0.7}$ | $92.3_{\pm0.9}$ | $\underline{92.9_{\pm0.5}}$ |
| Bal. Acc. | $59.66_{\pm1.7}$ | $66.2_{\pm1.1}$ | $66.9_{\pm0.9}$ | $73.3_{\pm1.7}$ | $73.6_{\pm2.1}$ | $75.3_{\pm1.8}$ | $\mathbf{77.7_{\pm1.1}}$ | $\underline{76.1_{\pm3.1}}$ |
| AuROC | $59.66_{\pm1.7}$ | $66.2_{\pm1.1}$ | $66.9_{\pm0.9}$ | $83.8_{\pm1.2}$ | $84.0_{\pm1.6}$ | $84.6_{\pm0.7}$ | $\mathbf{86.2_{\pm1.1}}$ | $\underline{85.2_{\pm0.9}}$ |
| AP | $27.50_{\pm3.0}$ | $17.6_{\pm0.9}$ | $40.1_{\pm1.6}$ | $56.0_{\pm4.1}$ | $57.1_{\pm4.5}$ | $61.4_{\pm2.5}$ | $\underline{64.8_{\pm3.8}}$ | $\mathbf{64.9_{\pm3.4}}$ |
| $F_1$ | $32.23_{\pm4.7}$ | $33.0_{\pm1.6}$ | $50.3_{\pm2.1}$ | $54.0_{\pm2.6}$ | $54.3_{\pm4.1}$ | $58.3_{\pm2.7}$ | $\mathbf{61.8_{\pm2.8}}$ | $\underline{61.4_{\pm4.0}}$ |
| MS | $19.36_{\pm3.4}$ | $50.6_{\pm2.2}$ | $33.8_{\pm1.9}$ | $51.1_{\pm4.3}$ | $51.9_{\pm3.8}$ | $54.4_{\pm4.0}$ | $\mathbf{59.1_{\pm2.0}}$ | $\underline{54.9_{\pm6.6}}$ |

## 5.2 QUANTITATIVE RESULTS

**MNIST-R classification.** We report classification results using selected baselines and our ContextFlow++ variants in Table 2. As expected the generalist model with image rotations in the data splits has 2.8 percentage points (p.p.) lower accuracy results because the same-size model without rotational invariance cannot be as successful in approximating larger data distribution. With the proposed ContextFlow++, we lower that accuracy gap to 1.0 p.p. The deterministic embedding-based method and variational dequantization variants have the highest performance metrics.

When compared to the conventional baseline [Lu and Huang, 2020] results, our context-conditioned variants improve classification accuracy by only 0.1-0.2 p.p. which signals about lack of useful general knowledge in MNIST-R. Another interpretation can be a relatively simple MNIST classification task with saturated accuracy metrics.

**CIFAR-10C classification.** Table 3 presents the same baselines but with very different outcome. First, overall accuracy is significantly lower (61.7%) and image corruptions increase accuracy gap between the models trained and evaluated on the undistorted CIFAR-10 and the corrupted CIFAR-10C to 6.6 p.p. (61.7% vs. 55.1%).

Second, the conventional conditioning approach is unable to surpass even the generalist model results. At the same time, the proposed ContextFlow++ converges well because the general knowledge is preserved in the fixed generalist parameters, where it leads to 2.6 p.p. (57.7% vs. 55.1%) higher accuracy. The best results are, again, achieved with deterministic embedding-based encoder and variational dequantization variants with one-hot binary context representation and $\arg\max$ method with $\log_2$ context compression.

**ATM failure prediction.** We reproduce Vargas et al. [2023] baselines in Table 4 using their public code but with the modified data splits. Particularly, we evaluate classic non-temporal machine learning methods: random forests [Breiman, 2001] and XGBoost [Chen and Guestrin, 2016]. The HYDRA model [Dempster et al., 2023] is a hybrid method with convolutional neural network (CNN) for feature extraction with temporal processing followed by the ridge classifier [Pedregosa et al., 2011].

We report flow-based generalist model and the best ContextFlow++ variants in Table 4. It is clear that even the generalist model significantly outperforms all baselines from Vargas et al. [2023] and our ContextFlow++ further improves failure prediction metrics. For example, ContextFlow++ with more advanced context encoders achieve the highest results and provides up to 6.3 p.p. additional $F_1$ score gain when compared to the generalist model.

Table 6: Unsupervised anomaly detection on real-world SMAP dataset with time series sensory data [Hundman et al., 2018]. Entity IDs (55) define the conditioning context. Unlike ContextFlow++ with a single model (# = 1), conventional baselines train and evaluate on a separate model for each entity (# = 55). Our ContextFlow++ significantly improves anomaly detection precision (P) and, hence, the $F_1$ score, while recall (R) and AuROC scores are saturated as in other baselines. The **best** and the second best metric's results, if metric is not saturated, are highlighted, %.

| Model | # | P | R | AuROC | $F_1$ |
|---|---|---|---|---|---|
| OmniAnom. | 55 | 81.30 | 94.19 | 98.89 | 87.28 |
| MTAD-GAT | 55 | 79.91 | 99.91 | 98.44 | 88.80 |
| CAE-M | 55 | 81.93 | 95.67 | 99.01 | 88.27 |
| GDN | 55 | 74.80 | 98.91 | 98.64 | 85.18 |
| TranAD | 55 | 80.43 | 99.99 | 99.21 | 89.15 |
| Generalist | 1 | 87.40 | 84.93 | 91.55 | 86.05 |
| ContextFlow++ | 1 | **88.64** | 99.19 | 98.66 | **93.62** |

To highlight the robustness of our probabilistic models, we conduct additional experiments where we subsample number of positive (failure) data points. Table 5 shows results where imbalance between positive and negative examples is increased from $10\times$ to $100\times$ by training set subsampling. With the subsampled ATM, we have two important observations. First, the gaps in metrics between the best ContextFlow++ models and other baselines increase by 1-2 p.p. Second, deterministic embedding-based approach does not perform as good as with MNIST-R, CIFAR-10C and the original ATM data. At the same time, the variational dequantization has the highest overall scores. Then, a fully-probabilistic model (including the context encoder) can be more robust when applied to real-world application settings.

**SMAP unsupervised anomaly detection.** We compare our models to popular baselines: OmniAnomaly [Su et al., 2019], MTAD-GAT [Zhao et al., 2020], CAE-M [Zhang et al., 2021], GDN [Deng and Hooi, 2021] and TranAD [Tuli et al., 2022]. It is common in these baselines to train and evaluate a separate model for each SMAP entity (# = 55). In contrast, our generalist model uses a single model for all entities, which leads to lower performance metrics in Table 6. Then, we finetune our variational ContextFlow++ variant with the context defined as a discrete entity ID. This allows to significantly improve generalist's metrics (7.6 p.p. gain in $F_1$ score w.r.t. the generalist result) and outperform the selected baselines. Our approach leads to a major drop in complexity since we train and keep all additive contexts in a single checkpoint and, additionally, our model learns the decoupled common generalist knowledge.

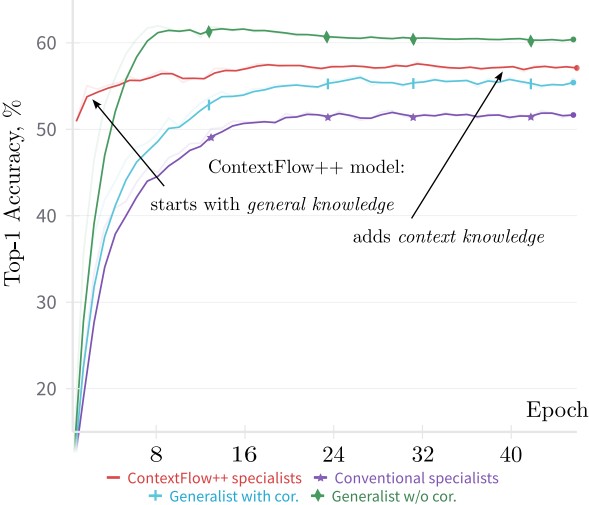

Figure 4: Top-1 accuracy of CIFAR-10C on test split vs. training epochs. Generalist model experiences significant accuracy drop when compared to the same model trained on the undistorted CIFAR-10. Our ContextFlow++ with `arg max`-based context encoder explicitly decouples general and context-specific knowledge. In comparison with conventional conditioning method, ours converges faster and results in higher accuracy metric on CIFAR-10C.

## 5.3 QUALITATIVE EXPERIMENTS

Figure 4 visually compares top-1 test-set accuracy for a subset of Table 3 models vs. training epochs. We plot accuracy of the generalist model trained and evaluated on the undistorted CIFAR-10 as well as corrupted CIFAR-10C. Also, we show our ContextFlow++ and conventional conditioning method [Lu and Huang, 2020] with exactly the same architectures and variational `arg max` context encoders.

Our ContextFlow++ approach has two main advantages as shown in Figure 4. First, it starts with the generalist knowledge encoded in its parameters which significantly increases convergence stability and speed. Second, the added context encoder allows to employ domain-specific knowledge and increase final performance metric. In practice, it can be useful when thousands and millions of contexts are encoded in multidimensional mixed-variable vectors.

## 5.4 COMPLEXITY ANALYSIS

Table 7 shows complexity estimates for the flow models from Table 3 that are applied to CIFAR-10C dataset. We report parameter count and latency on P100 GPU with mini-batch size of 256 during the training and evaluation phases.

The low-complexity context encoders with uniform dequantization and deterministic embeddings have comparable to

Table 7: Parameter count and P100 GPU latencies (batch size = 256) on CIFAR-10C. Results, reported as [Lu and Huang, 2020]→ContextFlow++, show that out method has lower parameter count and similar to the prior method latency. Ours $\arg\max$ variant with $\log_2$ context compression is preferable in terms of parameters-performance trade-off.

| Metrics → Method ↓ | Parameters, millions | Latency, ms | |
|---|---|---|---|
| | | Train | Eval |
| Generalist | 2.2 | 84 | 42 |
| Integer uniform | 3.5→ 2.9 | 138→141 | 55→71 |
| Integer $\arg\max$ | 4.1→ 4.0 | 402→403 | 194→185 |
| One-hot uniform | 4.0→ 3.3 | 155→152 | 74→ 81 |
| One-hot variational | 5.1→ 5.0 | 383→398 | 185→205 |
| Embed. determinist. | 6.2→ 5.6 | 140→145 | 55→ 69 |
| Embed. stochastic | 27.7→27.6 | 384→380 | 184→198 |

generalist model latencies (140 vs. 84 ms for training and 70 vs. 42 ms at evaluation), but can be very different in parameter count (2.9, 3.3 and 5.6 for ContextFlow++ variants vs. 2.2 millions for the generalist) depending on the context processing. At the same time, probabilistic context encoders with generative flow architecture have higher latency (400 vs. 84 ms for training and 200 vs. 42 ms at evaluation) and also variable parameter counts (4.0, 5.0 and 27.6 millions).

The parameter count for ContextFlow++ is lower than the conventional baseline [Lu and Huang, 2020] due to lack of concatenation that increases the dimensionality of internal vectors. At the same time, the latencies for both methods are very similar due to the nature of Table 1 operations. To conclude, the $\arg\max$ variant with the embedded $\log_2$ context compression can be a preferred method with further encoder architecture optimizations as a trade-off between complexity and promising performance gains in our experiments.

## 6  CONCLUSIONS

In this paper, we addressed the limitation of previous conditional normalizing flow models. Our additive contexts increased applicability of flow models to setups where flexible and accurate context-specific knowledge modeling is crucial. Then, we explored the related topic of enabling discrete variables in the conventional flow framework and proposed the mixed-variable ContextFlow++ architecture with additional generative flow-based context encoders.

Our experiments with supervised image classification, predictive maintenance and unsupervised anomaly detection showed advantages of our flow-based architecture with sampling from surjective context encoders followed by likelihood estimation using modified ContextFlow++ bijective decoder. We believe that this approach can be extended to recent ODE-type continuous flow architectures and other types of contextual information e.g., relational graphs.

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
