# OpenReview forum: "ContextFlow++: Generalist-Specialist Flow-based Generative Models with Mixed-variable Context Encoding"
_auai.org/UAI/2024/Conference — UAI 2024 poster_

### Official Review · Reviewer_gFu4 · 2024-03-18

**Q2-1 Originality-Novelty:** 3
**Q2-2 Correctness-Technical Quality:** 3
**Q2-5 Clarity Of Writing:** 3

**Q10 Ethical Concerns:**

no concerns

**Q1 Summary And Contributions:**

In order to support a practical setup where a set of context conditioned (specialist) models are trained with the fixed pretrained general-knowledge (generalist) model, the authors propose ContextFlow++ approach using an additive conditioning with explicit generalist-specialist knowledge decoupling.

Experiments on rotated MNIST-R, corrupted CIFAR-10C and real-world ATM predictive maintenance benchmarks show that ContextFlow++ converges faster and achieves higher performance metrics.

**Q2-3 Extent To Which Claims Are Supported By Evidence:**

3: Good: the main claims are supported by convincing evidence (in the form of adequate experimental evaluation, proofs, (pseudo-)code, references, assumptions).

**Q2-4 Reproducibility:**

3: Good: key resources (e.g. proofs, code, data) are available and key details (e.g. proofs, experimental setup) are sufficiently well-described for competent researchers to confidently reproduce the main results.

**Q3 Main Strengths:**

strong:
1. a general approach to support additive context-conditioning for the generalist-specialist setup in bijective normalizing flow transformations.
2. the addressing of mixed-variable input data and contexts that are common in practical applications using their proposed ContextFlow++ architecture.
3. extensive experiments on toy image classification and real-world predictive maintenance benchmarks show advantages of their ContextFlow++ approach.

**Q4 Main Weakness:**

weak:
1. for the same target tasks, prefer to learn more comparisons with other architectures besides flow-based methods, such as transformer architecture or diffusion models.

**Q5 Detailed Comments To The Authors:**

1. is it feasible to make a high-level comparison of your flow-based framework with other frameworks such as diffusion models and transformer architecture?
2. when it comes to decoupling specialist and generalist, will retrieve-based generation be helpful in your architecture? So the retriever collects general and special domain external knowledge and the generator refine its outputs accordingly.

**Q9 Complying With Reviewing Instructions:**

Yes

---

> ### Author Rebuttal · Authors · 2024-04-03
>
> We appreciate your feedback that recognizes advantages of our approach and anticipation of appealing future applications for the ContextFlow-type models. We would like to respond to your comments.
>
> **1. for the same target tasks, prefer to learn more comparisons with other architectures besides flow-based methods, such as transformer architecture or diffusion models.**
>
> Thank you for this suggestion. Let us elaborate on these topics in details to clarify our logic in the current paper structuring.
>
> - **Transformers for the flow-based models**.
>
> We actually support transformer-based bijections in our code (argument `---coupling trans`). We found they work better than the convolutional ones in larger-scale image domain such as CIFAR-10C at the expense of higher complexity. On the other hand, for other benchmarks such as failure prediction in time series, transformers tend to overfit to the training set due to smaller-scale data. Therefore, we decided to remove such architectural option from the experimental section due to the page limit and focus on the main objective *i.e.* the mixed-variable conditioning. We consider adding these experiments with the extended page limit in camera-ready version.
>
> - **Transformers for other (discriminative) models**.
>
> The main disadvantage of discriminative models (CNN-based or transformer-based) is that they can hardly support data setups with weaker supervision (heavy-imbalanced as in Table 5, semi-supervised or weakly-supervised). That is why even high-complexity transformer-based discriminative models cannot compete with probabilistic classifiers using a single-phase optimization. One of the possible solutions for them is to use sophisticated multi-phase training scheme as in Eldele et al. [1]. We already accomplished experiments for the semi-supervised setup (5% of labeled data) on the UCI HAR dataset to compare to various baselines in [1] and achieved similar or slightly better results without conditioning. Unfortunately, the context-conditioning (our main proposal) cannot be accomplished on HAR's test set because user IDs do not overlap in the training and test sets. So, the only option is to present results for the training and validation splits which is not ideal. Therefore, we decided to exclude HAR experiments in the submission to avoid further confusions and included only the benchmarks with feasible context-conditioning. Therefore, we consider including HAR's results in the camera-ready version with the discussion about data split limitations. In general, there are very few public datasets with additional context information that can be used in our experiments.
>
> - **Diffusion models**.
>
> Thanks for pointing to this. We mentioned VAEs and diffusion models in the Introduction. They are less flexible and not able to estimate data likelihoods exactly, which is a major disadvantage in failure prediction and multi-class classification (*i.e. the non-conditional part of ContextFlow*). That's why they are limited to certain types of applications e.g., reconstruction-based weakly-supervised anomaly detection. At the same time, we agree that the diffusion models shine in the sampling (*i.e. the conditional part of ContextFlow*). Therefore, recent research [2] shows that the diffusion models can be generalized by the flow-type framework (including infinitesimal continuous flows with ODEs, SDEs and flow-matching). Then, [2] allows to combine advantages of the flows and diffusion models which is the interesting future direction especially for high-dimensional data.
>
> **1. is it feasible to make a high-level comparison of your flow-based framework with other frameworks such as diffusion models and transformer architecture?**
>
> Sure, we can add additional description and comparison into our related work section.
>
> **2. when it comes to decoupling specialist and generalist, will retrieve-based generation be helpful in your architecture? So the retriever collects general and special domain external knowledge and the generator refine its outputs accordingly.**
>
> Yes, this is a great direction for future work. In fact, we work in this direction but for very large-scale data domains (e.g., language). As you pointed, in this work we proposed a *general approach* for a class of finite (fixed number of bijections) normalizing flows and evaluated it using relatively medium-scale datasets. However, the retrieve-based generation with millions of contexts is what we target the next. However, that would require substantial architectural changes as discussed above. We would be happy if you can suggest us relevant datasets or benchmarks for retrieve-based generation.
>
> [1] [Eldele et al. Self-supervised Contrastive Representation Learning for Semi-supervised Time-Series Classification. In TPAMI 2023.](https://arxiv.org/abs/2208.06616)
>
> [2] [Albergo et al. Stochastic Interpolants: A Unifying Framework for Flows and Diffusions. In arXiv:2303.08797, 2023](https://arxiv.org/abs/2303.08797)

---

### Official Review · Reviewer_bVab · 2024-03-22

**Q2-1 Originality-Novelty:** 2
**Q2-2 Correctness-Technical Quality:** 3
**Q2-5 Clarity Of Writing:** 3

**Q1 Summary And Contributions:**

- This paper proposes ContextFlow++, a method for taking a pre-trained "general" normalizing flow model and adding conditioning information via a learned "specialist" set of weights. The motivation is to allow for a general purpose model to be adapted to new conditioning variables unseen at testing time (similar to ControlNet for diffusion models).
- This is achieved by freezing a pre-trained generalist model, followed by adding a new context-dependent set of weights to the model.
- Since many conditioning variables are discrete, the authors carefully outline how to handle discrete variables through encoding or dequantization.
- Experiments on MNIST/CIFAR-10C and a real-world ATM failure detection dataset show the proposed method is competitive with existing normalizing flow based models.

**Q2-3 Extent To Which Claims Are Supported By Evidence:**

2: Fair: the main claims are somewhat supported by evidence (but the experimental evaluation may be weak, or does not match entirely with the claims, important baselines may be missing, proofs contain important ideas but lack rigor, algorithmic details are only discussed superficially, references are imprecise, assumptions are not sufficiently motivated or explicated, etc.).

**Q2-4 Reproducibility:**

3: Good: key resources (e.g. proofs, code, data) are available and key details (e.g. proofs, experimental setup) are sufficiently well-described for competent researchers to confidently reproduce the main results.

**Q3 Main Strengths:**

- The main idea of the paper, to allow for post-hoc conditioning of normalizing flows, is an interesting idea and potentially of interest to the community.
- The method itself (Equation 6) is a fairly simple modification of existing NF techniques, which makes it easy to implement into existing frameworks.
- Empirically, the proposed method performs as well as or better than the considered baseline (Table 2, 3) in the MNIST and CIFAR-10C benchmarks.

**Q4 Main Weakness:**

- In Table 4/5, I would have expected the authors to compare their proposed method against a flow-based model that was trained with the context (a "specialist" in the author's words) -- similar to what is done in Table 2/3. Currently, the authors only compare against some classical methods and a CNN, as well as a "generalist" flow-based method.
     - It is not too surprising that the ContextFlow model outperforms the "generalist" flow-based method (as it has access to more information).
     - The real question that should be determined is: does the ContextFlow approach help over simply training with the context available?
- In Figure 4 (and the abstract) the authors make the claim that "ContextFlow++ converges faster". I'm not sure that this is a fair claim -- in particular, claiming the red curve (Figure 4) converges the fastest is unfair as it is starting from a pretrained model. Similarly, claiming the green curve converges faster than the purple/blue curves is unfair as this model has access to uncorrupted data, unlike the purple/blue curves.
- Equation 5 assumes that the data $v$ and context $c$ are independent. This is a *very* strong assumption, and one that almost certainly does not hold in practice (if the data and context are independent, why would the $c$ be useful for predicting $v$?). Moreover, Equation 6 does not actually use this assumption, since $u$ depends (implicitly) on both $v$ and $c$.

**Q5 Detailed Comments To The Authors:**

See weakness section.

**Q9 Complying With Reviewing Instructions:**

Yes

---

> ### Author Rebuttal · Authors · 2024-04-05
>
> We appreciate your feedback that highlights important parts of this paper. Hopefully, the following responses can address your concerns.
>
> - **In Table 4/5, I would have expected the authors to compare their proposed method against a flow-based model that was trained with the context... similar to what is done in Table 2/3.**
>
> Thank you for bringing this point that we did not clearly mention. One of our unexpected experimental findings was that the *conventional context-conditioned models are quite unstable during training*. Particularly, such unstable behavior is amplified with more complex input data distributions and more sophisticated context encoding schemes (e.g., our generative context encoder). Therefore, unlike Table 3 results where only a subset of conventional baselines did not converge well, all of them diverge on ATM dataset. That's why we cannot report their results. We will explicitly mention the training instability on ATM dataset for the conventional approach.
>
> - **The real question... does the ContextFlow approach help over simply training with the context available?**
>
> Hypothetically, with the identical data and almost the same architecture both the conventional baseline and the ContextFlow should converge to the same results. Practically, we observe comparable results on *small-scale MNIST-R* task only. With larger-scale data with complex distributions in CIFAR-10C and ATM datasets, ContextFlow outperforms conventional method and, in addition, the latter often fails to converge.
>
> - **In Figure 4 (and the abstract) the authors make the claim that "ContextFlow++ converges faster"... is unfair as it is starting from a pretrained model.**
>
> We kindly disagree. The ContextFlow enables such pretraining and with the much higher latency for training with context  conditioning, our statement is fair. Let us quantify the training time for both methods *to reach the same accuracy* in Figure 4. ContextFlow approaches the baseline accuracy after approximately 1/3 of total number of epochs there. Then, using our wall-clock training time estimates below, ContextFlow converges to the same accuracy in almost 50% less time for the $\arg \max$ variant: (12.7 + 61.2/3) / 61.1 = 0.54.
>
> Therefore, our overall conclusion holds: as complexity of conditioning grows with more contexts, it takes less time to do pretraining and finetuning using our ContextFlow than the conventional procedure. We will add this table to quantitatively justify our findings.
>
> Mean training and evaluation latencies for one epoch using the default batch size of 256 on CIFAR-10 dataset:
>
> | Latency, sec | Train || Eval ||
> | ---         | :----:|:----:| :----:|:----:|
> | Methods | Conventional  | ContextFlow | Conventional | ContextFlow |
> | Generalist              |    12.7 | - | 5.8         | - |
> | Integer uniform         |    16.4 |  16.2 |  7.5 |  8.2 |
> | Integer $\arg \max$     |    61.1 |  61.2 | 27.3 | 27.5 |
> | One-hot uniform         |    17.8 |  17.8 |  9.1 |  9.6 |
> | One-hot variational     |    62.8 |  17.8 | 27.3 | 27.6 |
> | Embedding deterministic |    17.8 |  41.0 |  7.4 |  8.1 |
> | Embedding stochastic    |    60.8 |  61.9 | 26.2 | 26.7 |
>
> - **Equation 5 assumes that the data  and context  are independent. This is a very strong assumption... Moreover, Equation 6 does not actually use this assumption, since u depends (implicitly) on both $v$ and $c$.**
>
> Thank you for this comment. Indeed, this is one of the *key parts of this paper* which requires detailed discussion which we avoided due to the page limit. Only the masked autoregressive flows (MAFs) (Section 3.1 in [1]) are *true universal approximators between any two distributions* using a *single large-enough bijection* (Section 2.2 in [1]). In this sense, none of the used bijections in our experiments model data vectors without context vectors $(p(v))$ or with them $(p(v,c))$ as a true joint density from Equations (1,4,5) *in a single layer*, where an autoregressive bijection decomposes joint density as $p(v,c) = \prod_{i=1} p(z_i | z_{<i})$ and $z = \[v,c\]$. Instead, it is presumed that a sufficient number of bijection layers with vector permutations lead to an accurate to $p(v,c)$ estimate. In practice, non-autoregressive bijections have comparable to MAFs likelihood estimates without slow sequential sampling.
>
> Our equation (6) generalizes widely-used bijections using expression for a single layer. It contains the binary matrix **M** that can implement all types of masks with the exception of the full MAF mask for $p(v,c)$ above. Instead, we are limited to $p(v,c) = p(v) p(c) = \prod_{i=1} p(v_i | v_{<i}) \prod_{j=1} p(c_j | c_{<j})$. So, the $v$ and $c$ independence are mentioned regarding a single bijection (not the layered architecture as you noticed for the layer's output $u$), which is a common assumption. We will add this discussion and clarify our Equation (5).
>
> [1] Papamakarios et al. Normalizing flows for probabilistic modeling and inference. In JMLR, 2021

---

### Official Review · Reviewer_gyaN · 2024-03-24

**Q2-1 Originality-Novelty:** 3
**Q2-2 Correctness-Technical Quality:** 3
**Q2-5 Clarity Of Writing:** 3

**Q1 Summary And Contributions:**

The paper studies the problem of training conditional normalizing flows (specialists) with a low computation cost, given a pretrained unconditional flow (generalist). An efficient architecture that learns conditional additive corrections to the unconditional neural nets in each coupling layer is proposed, and various ways to use it in discrete data settings are discussed. This architecture is tested on MNIST and CIFAR-10 image classification (conditioning on the type of image corruption) and an ATM failure time series dataset.

**Q2-3 Extent To Which Claims Are Supported By Evidence:**

2: Fair: the main claims are somewhat supported by evidence (but the experimental evaluation may be weak, or does not match entirely with the claims, important baselines may be missing, proofs contain important ideas but lack rigor, algorithmic details are only discussed superficially, references are imprecise, assumptions are not sufficiently motivated or explicated, etc.).

**Q2-4 Reproducibility:**

3: Good: key resources (e.g. proofs, code, data) are available and key details (e.g. proofs, experimental setup) are sufficiently well-described for competent researchers to confidently reproduce the main results.

**Q3 Main Strengths:**

- Clear presentation of the preliminaries and proposed method, including good figures. It is easy to understand even for someone who does not have much experience in designing NF architectures.
- The experimental setup is effective at verifying the hypotheses about the expressivity of the proposed additive conditioning, since training the context conditioning model while keeping other components fixed is as effective as training a context-conditioned model end-to-end.
- Code is provided (I did not run it, but it is clean and readable).

**Q4 Main Weakness:**

The main weakness: I am concerned that the experiments are not actually showing the hypothesized (or implicitly hypothesized) improvements that should come from the proposed conditioning method.
- What are **all** differences in architecture between the baseline generalist NF and the proposed conditional one?
  - In Figure 4, the ContextFlow++ model seems starting at a value that is already higher than that of the generalist model (though lower than the trained conventional conditional one), but the conditional models have not undergone any training. This means that the changes in architecture may be playing a role.
- Computation cost comparison in Table 8 is not comparing the other kinds of conditional NF with the proposed one. It is only comparing unconditional NF with the proposed conditional one.
  - It is also not showing training cost. If we are fixing the generalist and only training conditional corrections, we should hope that the overhead is low (cost per iteration much less than that for the fully conditioned model trained from scratch).
- What happens if you do not fix the generalist model when training the context conditioning modules? How much more can be gained over the last rows of Tables 2 and 3?

**Q5 Detailed Comments To The Authors:**

Minor:
- First line of the paper: probabilistic generative models were not introduced by Kingma et al. in 2014!
- In Figure 4, please use markers and line styles and not just colours to distinguish the curves.

**Q9 Complying With Reviewing Instructions:**

Yes

---

> ### Author Rebuttal · Authors · 2024-04-03
>
> - **...the experiments are not actually showing the hypothesized... improvements**
>
> We appreciate your detailed feedback which is valuable to improve paper's readability. We would like to clarify your concerns using the explanations and experiments below.
>
> - **...differences in architecture between the baseline generalist NF and the proposed conditional one?**
>
> There is no difference between the generalist NF and the non-conditional part of the ContextFlow: they share the same parameters to preserve *general knowledge*. However, the ContextFlow's conditional part is implemented as a separate flow-based model (*i.e.* the context encoder in Figure 3) that adds the *context-specific knowledge*. The context encoder *operates very differently*. Unlike the non-conditional NF with likelihood estimation, the conditional one learns a context-conditioned distribution $q_{\lambda_c}( c | x_c)$ followed by the encoder bijections, where the vector $x_c$ can contain discrete variables. That's why its order of operations is reversed in Figure 3 (right). Then, we *sample context-conditioned continuous variables* from the context encoder and add the sampled vectors to the internal vectors with *general knowledge*.
>
> - **In Figure 4, the ContextFlow++ model seems starting at a value that is already higher than that of the generalist model...**
>
> Figure 4 shows results after the first training epoch (not before it). Otherwise, it was hard to fit other curves where y-scale starts from 0% accuracy to the page limit. ContextFlow begins exactly with the last generalist results and shows a minor gain even after the first epoch. That's why it is visually a bit higher w.r.t. to the generalist result. We will add the starting point accuracy to better illustrate this behavior.
>
> - **Computation cost comparison in Table 8 is not comparing the other kinds of conditional NF...**
>
> Thank you for pointing to this. We will add a note that the *conventional conditional NF architecture has marginally higher number of parameters* when compared to ours. This is because the conventional approach concatenates internal and condition vectors which results in the $(D+C)$-dimensional representations while ours are $D$-dimensional only. We compare the latency metric in the table below.
>
> - **It is also not showing training cost...**
>
> We report training and evaluation latencies when using the default batch size of 256 (unlike batch size of 1 in Table 8 for "edge" deployment) on ATM dataset. ContextFlow results in approximately the same latency (within the error bar) as the conventional method. There is a minor additional processing in certain layers (e.g., Conv$^{-1}_{1 \times 1}$ from Table 1), but the coupling layer is marginally less complex as explained above. Note that the conventional conditioning fails to converge on ATM and CIFAR-10C datasets (that is the reason we excluded the conventional baseline in Tables 4-5).
>
> | Latency, ms | Train || Eval ||
> | ---         | :----:|:----:| :----:|:----:|
> | Methods | Conventional  | ContextFlow | Conventional | ContextFlow |
> | Generalist              |    29.6 |-|        16.9        |-|
> | Integer uniform         |    32.2 |  32.9 | 18.5 | 18.6 |
> | Integer $\arg \max$     |   124.5 | 122.8 | 57.8 | 55.0 |
> | One-hot uniform         |    35.5 |  36.3 | 20.6 | 20.3 |
> | One-hot variational     |   129.9 | 131.1 | 59.6 | 58.8 |
> | Embedding deterministic |   36.8  |  41.0 | 17.5 | 19.4 |
> | Embedding stochastic    |   128.6 | 128.2 | 57.1 | 57.1 |
>
> - **What happens if you do not fix the generalist model when training the context conditioning modules? How much more can be gained over the last rows of Tables 2 and 3?**
>
> If we do not fix generalist parameters, ContextFlow variant with the deterministic learned embedding shows:
>
> Generalist -> fixed -> all trainable
>
> MNIST-R (top-1 acc.) : 95.35 -> *97.23* -> **97.64**
>
> CIFAR-10C (top-1 acc.) : 54.27 -> *57.37* -> **58.08**
>
> ATM  (F$_1$ score) : 40.87 -> **41.58** -> *40.98*
>
> We can see a minor improvement in image classification accuracy. Most likely, this is a result of a semantic overlap in the data: digit "8" is the same with 180 degree rotation in MNIST-R and a small difference between a corruption with severity level 1 and 2 in CIFAR-10C *etc.* ATM machines overlap less and the fixed generalist parameters lead to higher F$_1$ score. Practically, we envisioned the generalist training with large-scale data followed by the finetuning with small-scale data but with millions of contexts. Then, the ContextFlow with fixed general knowledge can be preferable due to complexity/optimization reasons and the unseen yet contexts.
>
> - **...probabilistic generative models were not introduced by Kingma et al.**
>
> We agree with this. We referred to this paper because it has a comprehensive overview of previous papers and the semi-supervised setup similar to ours.
>
> - **In Figure 4, please use markers and line styles...**
>
> Thank you for pointing to the line style. We will fix it.

---

### Meta-Review · Area_Chair_BBGp · 2024-04-15

This paper proposes ContextFlow++, a method for taking a pre-trained "general" normalizing flow model and adding conditioning information via a learned "specialist" set of weights.

During the reviewing process, this paper was reviewed by three expert reviewers. Two of them support this paper, while one reviewer gave a borderline reject rating.

In the rebuttal period, the authors provided detailed feedback for the questions raised in the reviews. Most of the concerns are addressed by the rebuttal, especially the concerns from Reviewer gyaN, who gave "borderline reject". The authors are required to include the discussions and additional results during the rebuttal period in the final version.